# Advances in the Clinical Application of Platelet-Rich Plasma in the Foot and Ankle: A Review

**DOI:** 10.3390/jcm12031002

**Published:** 2023-01-28

**Authors:** Djandan Tadum Arthur Vithran, Miao He, Wenqing Xie, Anko Elijah Essien, Michael Opoku, Yusheng Li

**Affiliations:** 1Department of Orthopaedics, Xiangya Hospital, Central South University, Changsha 410008, China; 2National Clinical Research Center for Geriatric Disorders, Xiangya Hospital, Central South University, Changsha 410008, China

**Keywords:** platelet-rich plasma, foot and ankle, diabetic foot ulcers, plantar fasciitis, Achilles tendon pathology, ankle osteoarthritis, review

## Abstract

Autologous and recombinant biologic substances have been generated as a result of the research into the cellular features of the healing process. Orthobiologics are increasingly being used in sports medicine and musculoskeletal surgery. Nevertheless, clinical data are limited; consequently, further studies are required, particularly in foot and ankle pathologies. This review aims to provide evidence of the most recent literature results and ignite the interest of orthopedic specialists eager for an update about the most current discussion on platelet-rich plasma (PRP) clinical applications in the foot and ankle fields. Previous studies have shown that platelet-rich plasma can be beneficial in treating various conditions, such as chronic foot ulcers, osteoarthritis, Achilles tendinopathy, etc. Despite the positive effects of PRP on various musculoskeletal conditions, more prospective studies are needed to confirm its effectiveness at treating ankle and foot pathologies. In addition to clinical trials, other factors, such as the quality of the research and the procedures involved, must be considered before they can be used in patients. More long-term evaluations are needed to support or oppose its application in treating foot and ankle disorders. We present the most extensive review of PRP’s clinical applications in the foot and ankle field.

## 1. Introduction

In recent decades, with advances in basic medical science, we know that platelets have multiple physiological functions. In 1978, in exploring the pathogenesis of atherosclerosis, it was found that 10% serum could significantly promote the proliferation of smooth muscle cells in in vitro experiments, but this effect of promoting the cell proliferation disappeared after the replacement of the platelet-poor serum [1]. Witte first discovered a platelet-derived growth factor in platelet-alpha granules in 1978 (platelet-derived growth factor, PDGF) [2]. Over the next 20 years, platelets were found to contain the transforming growth factor β (TGF-β), insulin-like growth factor (IGF), epidermal growth factor (EGF), vascular endothelial growth factor (VEGF), fibroblast growth factor (FGFs), etc. [3,4]. Since the 1990s, with the rise of translational medicine worldwide, platelet-rich plasma (PRP) has gradually been used clinically. The PRP separation and preparation process is simple and can be obtained only after venous puncture and centrifugation before the application, which is almost non-invasive [5]. Since it is isolated from autologous blood, no immune response will be generated during the application. In 1998, Marx first applied PRP in the clinical repair of mandibular defects and found that PRP could significantly shorten the osteogenic repair process [6]. Since then, PRP has gradually been used in orthopedic surgery to promote bone fusion and fracture repair, and to accelerate soft tissue repair in acute and chronic tendon injuries [7,8]. This has also attracted much attention in the field of foot and ankle surgery because of multiple foot and ankle disorders, such as Achilles tendon diseases, adult-acquired flatfoot deformity, ankle fractures, ankle sprains, midfoot arthritis, osteochondral defect of the talus, and plantar fasciitis, which can severely affect patients’ daily lives and is usually treated conservatively. The surgical treatment for foot and ankle conditions can lead to long-term complications and increase the patient’s morbidity. The estimated cost of foot and ankle surgery for Medicare patients in 2011 was $11 billion, a 38% increase from the previous decade [9]. Foot and ankle surgery has the highest complication rate and may be associated with articular cartilage injury; wound complications; instrument breakage; infection; nerve, tendon, and ligament injury; and long-term nerve damage [10,11]. Biological therapies have become more prevalent in treating foot and refractory ankle conditions. PRP is most commonly used in the outpatient department [12,13]. Due to the increasing popularity of PRP among professional athletes and the media (Figure 1 and Figure 2), the global market for this product has significantly risen. According to researchers, the market will be worth $451.9 million by 2024 [14].

Additionally, various molecules and features of PRP, such as antibacterial, analgesic, and anti-inflammatory actions, may be advantageous for bone tissue regeneration [15,16,17]. Moreover, it has been suggested that platelets may activate peripheral blood mononuclear cells (PBMC), which then secrete IL-10, an anti-inflammatory cytokine implicated in tissue regeneration [18]. Leukocyte-rich PRP and leukocyte-poor PRP have been the subject of unresolved disputes for the past several years. However, these and other aspects must be considered when determining a PRP product’s optimal biological activity [6]. Despite PRP’s beneficial effects on tissue regeneration, its efficacy in bone healing remains debatable [15,19,20,21,22]. In vitro studies have demonstrated a dose-dependent effect of PRP on osteoblasts and fibroblast differentiation, with the best outcomes obtained with a low platelet concentration as opposed to a high one [15,20]. In contrast, the results obtained from in vivo and clinical research are contested, and with variable platelet concentrations of PRP employed and different protocols, the kind of bone defect and different animal species have been suggested as possible causes [15,21,23,24,25]. PRP utilized as an adjuvant to bone grafts was reported to have a favorable effect on the treatment of periodontal intraosseous deficiencies; however, it appeared to be ineffective at increasing bone growth in sinus lift treatments [22,26]. In contrast, PRP injection proved beneficial in surgery for patients with delayed bone union and nonunion [1].

One of the latest reviews in foot and ankle pathology to evaluate the clinical application of PRP was conducted in 2018. The study revealed no significant differences in the effectiveness of PRP compared with other procedures when treating acute Achilles tendon ruptures. The authors of the study also noted that additional studies are required to confirm the efficacy of this treatment. The studies on using PRP to treat chronic tendinitis revealed no significant effects on the procedure’s effectiveness [27]. However, they did not provide sufficient evidence supporting this treatment’s use and highlighted the lack of evidence supporting its use in treating other conditions. Numerous clinical studies show that PRP can help heal soft tissues and bones. However, to our surprise, few properly designed scientific research are available, and its use in ankle and foot pathologies is still unclear. This review aimed to investigate recent studies using platelet-rich plasma therapy in surgical and non-surgical foot and ankle-related conditions, wound healing, and diabetic-related issues to bring light to the topic for surgeons and physicians of these fields.

## 2. Platelet-Rich Plasma Preparation

Platelet-rich plasma (PRP) is a type of blood product commonly used to treat acute and chronic musculoskeletal conditions that do not expose patients to immune reactions. This product contains over 1500 growth factors and cytokines, which can affect the development of various cell forms, such as blood vessels and stem cells [28,29]. It has a high concentration of various growth factors, such as insulin-like, vascular, and fibroblast growth factors (Table 1). These factors can affect the development of specific cell types, which can be beneficial for the healing process of tissues [8,30]. The global market for platelet-rich plasma is expected to grow steadily over the next few years. This product can treat various musculoskeletal conditions, such as tendon injuries [31].

Various methods can be used for PRP production, but all of them have one thing in common: they are extracted from the blood that has already been treated with anticoagulants. They are then processed for up to an hour before being injected into the injured tissue.

There are significant differences in formulation and generation between commercial systems [44,45]. Variations in the platelet concentration, leukocyte concentration, growth factor content, and differences in isolation and activation procedures are all variables [46,47,48]. Due to the varying characteristics of PRP, it has been complex to compare the available literature on this biological substance. Recently, various studies have been conducted on the role of the concentration of leukocytes in the composition of PRP [49].

Currently, the market for PRP is segmented into two categories: leukocyte-rich and leukocyte-poor. According to studies, the presence of leukocytes in the product can contribute to the accumulation of specific inflammatory mediators, such as IL-1, IL-6, TNF, and IL-8 [50,51]. Due to the pro-inflammatory effects of leukocytes, many studies investigating the use of PRP for tendinopathy have shown that leukocyte-reduced PRP is superior to leukocyte-rich PRP [28,52,53,54]. Leukocyte-rich PRP formulations are known to have unique benefits. They can help support the natural inflammatory response needed for the healing process [55]. It has been hypothesized that leukocyte-reduced PRP is more effective and safer for innate cells in intra-articular applications [12,55].

The increasing number of orthopedic conditions treated using PRP has led to the growth of the global market for this product over the past two decades [56]. Despite the lack of definitive evidence supporting the use of PRP for various orthopedic conditions, the media has portrayed it as an effective treatment for athletes. This has led to the widespread popularity of this product among highly-trained athletes [57]. According to Kantrowitz et al., the team physicians’ decision to use PRP was influenced by feedback from their patients [56].

Although PRP has been widely promoted, it still has a long way to go before it can be considered a standard treatment for orthopedic conditions. Aside from the composition of the PRP, other factors, such as the timing of the injection and the number of injections, are also taken into account to determine its effectiveness [58].

## 3. PRP Applications in Foot and Ankle

The PRP therapy technique is still relatively new, yet it has already seen widespread application in the orthopedic field. Although numerous clinical and basic research have revealed that PRP can improve the healing of bone and soft tissue, its therapeutic usefulness in the field of foot and ankle surgery is still controversial due to limited clinical application data.

### 3.1. Effectiveness of PRP for Bone Nonunion

PRP’s influence on bone healing has been extensively studied in vitro and in vivo [59,60,61,62,63,64,65]. The hypothesis is that platelets and their growth factors will boost osteopontin, osteoprotegerin, osteoblast, osteoclast-like cells, and the differentiation of myoblasts and osteoblastic cells [63,64,65,66,67]. PRP’s effectiveness at bone healing is still debated. Many studies indicate promise, while others show little difference between PRP and control or standard products.

In a study conducted by Gandi et al. [68], nine patients with nonunion after surgery for foot and ankle fractures were treated with PRP. All these patients underwent the initial surgery within 20 days of the fracture and were diagnosed with nonunion within four to ten months after surgery. PRP combined with autograft was applied to the nonunion in the second revision surgery. The results showed that all nonunions healed after revision, and the mean healing time was 60 days. The authors also compared the growth factor concentrations in the hematoma at the fracture site in patients with nonunion and union and found that the concentrations of PDGF and TGF-β in nonunion hematoma were significantly lower than those in fresh fractures. This study suggests that applying PRP in the nonunion bone site and releasing growth factors after platelet activation may play a key role in promoting bone healing [68].

In a prospective clinical study by Bibbo, 62 patients with high-risk factors for nonunion (Table 2) for elective foot and ankle surgery were followed for six months after receiving PRP [69]. The patients underwent surgery on different parts of the foot and ankle. Some of the patients received PRP therapy and autologous bone graft as required. The efficacy of PRP was evaluated by radiography every two weeks after surgery, and 94% of patients achieved bone union on average 41 days after surgery. The mean bone healing time of patients treated with PRP alone was 40 days, while that of patients treated with combination therapy was 45 days [69]. The authors believe that PRP is important for treating patients at high risk of nonunion. However, there were limitations to this study. One was that these patients have different foot and ankle diseases, and the surgical methods they receive. Second, the study lacked a control group that did not receive PRP.

Coetzee and colleagues compared the effect of the PRP treatment with or without ankle replacement on the rate of syndesmosis fusion [70]. After the distal tibia and talus osteotomy, PRP was applied to the lower tibiofibular joint, the talus osteotomy’s surface, and the joint prosthesis’s surface. PRP and autograft were used in the lower tibiofibular joint. Radiographs were reviewed regularly after surgery. If the bone fusion is suspicious, they gave a CT review. The results showed that, compared with the 112 patients in the control group who did not receive the PRP treatment, the improvement rates of the lower tibiofibular fusion at 8 weeks and 12 weeks after surgery were 61.4% and 73.6%, respectively. Compared with the control group, the fusion improvement rates in the combined PRP and autograft group were 76% and 93.9%, respectively. PRP also significantly reduced the incidence of poor union or nonunion at the fusion site six months after surgery. Thus, what was said above, we can conclude that PRP application in ankle bone nonunion, although showing promising results, requires further research with a specific control group and a standard PRP formulation and application method.

### 3.2. Effectiveness of PRP Use in Ankle Sprains

Although there is currently limited evidence supporting the use of PRP in treating acute ankle sprains, it is becoming increasingly apparent that this treatment can improve the return to activity and reduce the severity of the injury. A small study on a group of elite athletes revealed that they had a quicker recovery and less pain after using PRP. In this study, the athletes who received ultrasound-guided PRP injections had a quicker recovery than those who received the same rehabilitation program without any treatment. They also performed better when returning to sports [71]. In a similar study, rugby players who suffered from syndesmotic injuries were more likely to recover faster after a PRP injection than those who had undergone the same rehabilitation program [72]. There is currently insufficient evidence supporting platelet-rich plasma (PRP) in treating ankle sprains. Rowden et al. conducted a double-blinded study to compare the effectiveness of the ultrasound-guided treatment of acute ankle sprain with local anesthetic versus standard saline injection [73]. They found that there was no statistical difference between the groups when it came to the VAS pain score and the Lower Extremity Functional Scale (LEFS). More research is needed to determine if this treatment can improve recovery and prevent further injury.

### 3.3. PRP Use in Achilles Tendon Pathology

#### 3.3.1. Achilles Tendinopathy an Overview

Achilles tendinopathy is an aseptic inflammation of the Achilles tendon that occurs in athletes and dancers, and chronic Achilles tendinitis is an aseptic inflammatory disease of the Achilles tendon that occurs in athletes, dancers, and sports enthusiasts [74,75].

Due to the increasing number of studies showing the effectiveness of PRP at treating various conditions, such as tendinopathy, the use of this product in clinical trials has increased [76].

Over the past decade, numerous clinical trials have shown that using PRP in treating tendinopathy can be effective [77,78,79,80]. Although the use of PRP in treating tendinopathy can be beneficial, the results of clinical trials are not always consistent [81,82,83,84,85,86]. For instance, some studies do not follow a standard procedure.

One of the main issues that still need to be addressed regarding the use of PRP in treating tendinopathy is the product’s efficacy due to the various factors that affect the patient’s condition [71]. Aside from the product’s efficacy, other factors that need to be considered when assessing a clinical trial’s effectiveness are the inclusion and exclusion criteria and the long-term follow-up [87,88].

#### 3.3.2. Effectiveness of PRP Injections in Nonoperative Management of Achilles Tendinopathy

The effectiveness of PRP at treating Achilles tendinopathy has been studied. Various treatment options are available, such as dry needling and shock wave therapy. Previous studies have shown that using PRP can stimulate the differentiation of tissue stem cells (TSCs) into tenocytes. However, it cannot reverse the differentiation of these cells into non-tendinous tissues [89]. A study conducted on using ultrasound-guided tenotomy followed by the injection of PRP to treat chronic tendinopathy revealed that this procedure is very safe and effective [90]. In 2019, a randomized controlled trial revealed that the use of PRP significantly reduced pain and increased the function of the patients [91]. Erroi et al. performed a study on 45 individuals with insertional AT. They examined the effectiveness of shock wave therapy and PRP injections. Although both treatment options improved measured outcomes, there were no significant differences between the groups [92].

A second study analyzed the effectiveness of PRP and dry needling at treating AT. It involved 46 participants. At six months, no significant difference was found between the two treatment methods [93]. The results of the studies suggest that the use of PRP in treating AT is either inferior or inadequate compared with other conservative procedures. The literature supporting the use of this treatment modality has been more consistent. Zhu Junshan et al. [94] and Zou Guoyou et al. [95] treated 15 and 11 patients with Achilles tendinitis with local PRP injections, respectively. Painful local injections of PRP were performed in the treatment of 15 and 11 patients with chronic Achilles tendinitis, and the patients were followed up for 18 months after the treatment. After 18 months of follow-up, magnetic resonance imaging (MRI) showed a significant improvement in the soft tissue inflammation around the Achilles tendinitis. The patients regained their normal gait and daily activities.

Filardo et al. [86] studied 27 patients (men and women) with chronic Achilles tendinitis. The average follow-up time was 54.1 months (30 months), and the results showed that the Victorian Institute of Sport Assessment (VISA-A), Visual Analogue Scalar Score (EQ-VAS), and Tegner motor level scores were significantly improved, and PRP injections for chronic Achilles tendinitis had a stable medium-term outcome. In a randomized, double-blind prospective study by Boesen et al., 60 men with chronic mid-Achilles tendinitis were treated with PRP, and the results showed that centrifugal training, combined with high-dose steroid or local anesthetic injections and PRP injections, were effective at reducing pain, improving motion levels, and reducing tendon thickness; however, drugs were more effective than PRP at improving chronic mid-Achilles tendinitis in the short term [96].

The study by Liu et al. compared the effectiveness of PRP injection patients and assessed the VISA-A score for 12 weeks, 24 weeks, and one year. It did not find a difference between the two groups. The PRP cohort showed an improvement in efficacy after six weeks, and the tendon thickness and pain scores of those treated with the treatment were significantly increased [97].

Hanisch et al. conducted a further study to compare the effectiveness of the two types of PRP injection at treating chronic AT. They analyzed the data of 84 patients who had previously failed conservative therapy. They found that using LR-PRP or LP-PRP did not result in significant differences [98].

Zhang et al. recently reviewed combined data from four randomized controlled trials in a systematic review and meta-analysis. In the included studies, there were no statistically significant differences between the PRP and saline groups in the Victoria Exercise Assessment of Achilles tendon (VISA-A) ratings, ultrasound measurements of tendon thickness, or ultrasound color Doppler activity [96,99,100,101,102].

In 2021, another study conducted by Kearny and colleagues revealed that using PRP injections was not as effective as using a dry needle for treating AT. The study revealed that using a single intra-tendinous injection of PRP did not reduce the symptoms of AT in the participants at six months [103]. Although the findings of this study do not support the use of PRP for treating AT, the authors have raised questions about the study’s methodology and participant choice. Clinically, the evidence supporting the use of PRP in treating AT does not support a conservative option. The lack of research on non-insertional and insertional AT also prevents further conclusions from being made regarding this issue [104,105].

#### 3.3.3. Effectiveness of PRP Injections in Surgical Augmentation in the AT

Various surgical procedures can be performed to treat AT. These include using a minimally invasive technique to debride the tendon and a percutaneous needle tenotomy procedure.

The study by Therman et al. analyzed the effects of debridement on 36 patients with midportion AT. They were randomized to the conventional technique or intraoperative PRP in combination with the procedure. It was found that the added PRP did not improve the outcomes compared with the debridement alone [106].

The study, which was conducted by Kirschner et al., analyzed the effectiveness of the two surgical procedures at treating chronic Achilles tendinosis. They were divided into two groups: the first was treated with the conventional technique, and the second was treated with PRP. After six weeks, the researchers found that the patients treated with PNT had lower pain scores than those treated with PRP [107]. However, the study’s results did not support using PRP as an augmentation to the tenotomy procedure.

Although there has been a positive result in the intraoperative use of PRP in Achilles tendinopathy, the present result is insufficient to claim its safe efficacy, but more research should be conducted on the topic.

### 3.4. Efficacy of PRP Injections in Achilles Tendon Rupture

The Achilles tendon rupture presents a new set of problems for the orthopedic surgeon compared to chronic tendinopathy. Treatment options include non-surgical, minimally invasive, and open surgery. The in vitro benefits of PRP make it a compelling treatment option for tendon and wound healing, in addition to surgical and non-surgical treatments. The use of whole blood and PRP for treating ruptured tendons has been associated with mixed clinical results. This is because the various application modalities and the biological composition of PRP can affect the results [100,108,109,110,111,112,113,114,115].

Gosens et al. randomized 20 patients with acute Achilles tendon ruptures into PRP treatment or control groups for surgical and non-surgical treatment groups [114]. The Achilles Tendon Rupture Score (ATRS), VISA-A, Foot and Ankle Outcome Score (FAOS), and Functional Ultrasound Elastography (FUSE) were used to monitor patients during weeks 1, 3, 6, 12, and 24. The PRP group showed significant improvements in ATRS, VISA-A, and FAOS scores, and the FUSE scans showed larger and stronger tendons.

In order to analyze the effectiveness of PRP at treating acute ATR, Keene et al. conducted a randomized, placebo-controlled trial. The study involved 230 participants. The researchers noted that using PRP in treating acute ATR was not associated with significantly improving the patient’s quality of life or functional outcomes [113]. The studies’ results suggest that using PRP to treat acute ATR does not improve clinical outcomes [113]. This therapy could help strengthen the healing process following an operation on the tendon.

A meta-analysis by Fitzpatrick et al. [116] analyzed the various studies that examined the effectiveness of PRP at treating tendinopathies. They found varying levels of blood-derived products used to treat these conditions. These included autologous whole blood, autologous conditioned serum, and leukocyte-poor PRP. Various preparation methods are utilized to produce PRP products [117]. The main factor that sets them apart from peripheral blood is their concentration. The methods used to produce PRP products contain different proportions of white blood cells and erythrocytes. This factor affects the therapeutic properties of the product and its biological composition [117,118,119].

De Carli et al. compared the effects of PRP injections in 30 individuals who had their Achilles tendon rupture surgically repaired [120]. At six months post-operative, the signal enhancement was lower in the PRP group than in the control group, indicating better tendon remodeling but no clinical changes.

The groups had no clinical differences between the functional tests or VAS, FAOS, or VISA-A scores. Regarding elasticity and functional outcomes, Schepull et al. found that PRP did not affect acute Achilles tendon healing. However, their findings are difficult to interpret due to significant patient variation.

A single-blind study was conducted on 30 individuals who had undergone a surgical procedure to repair an injured tendon. Schepull et al. [112] found that there was no biomechanical benefit from using 10 mL of PRP in treating the ruptured tendons. Instead, they applied a concentrate containing high levels of PRP to the site of the injury [112]. The researchers found no evidence of a biomechanical advantage from using 10 mL of PRP to treat the ruptured tendons. They also found that the concentration of PRP in the concentrate was 17 times higher than that of the patient’s peripheral blood [117,121,122,123].

A study by Alviti et al. [111] revealed that using the LR-PRP matrix over the site of the ruptured tendon significantly improved the ankle’s function. The study also reported that the patients who were treated with PRP augmentation had a significant improvement in their ankle motion efficiency.

A systematic review of the eight studies was conducted to analyze the data. These studies were conducted on 543 patients with a diagnosis of acute ATR. The authors identified five studies that analyzed the various types of PRP used in treating acute ATR. Only one study yielded significant positive results, revealing that the patients could recover a normal range of motion within four weeks following the injection. The results of the studies revealed that the use of PRP in treating acute ATR did not improve strength or functional outcomes. The authors concluded that the current evidence does not support the use of this therapy in this condition [124].

In 2016, a study conducted by Zou et al. revealed that using PRP as an adjunct to surgery for treating acute ATR could be beneficial. They divided the participants into two groups: the control group and the PRP group. At the three-month mark, the researchers noted that the PRP group exhibited better isokinetic muscle strength and improved Leppilah scores. The study’s results revealed that using PRP in treating acute ATR improved the ankle range of motion after two years [125]. Although this initial proof of its effectiveness is encouraging, further studies are needed to determine if it can help improve the healing process following an operation on the tendon. Table 1 provides a summary of the study’s quality and protocol.

Keene et al. also conducted a randomized controlled trial [113] to use 4 mL of LR-PRP in treating patients with acutely ruptured tendons. They found that this method prevented the infiltration of local anesthetics into the affected area. Despite the positive results of the laboratory studies, the authors concluded that the use of PRP did not appear to have a detectable effect on the healing of the injured tendons [113,118].

The results of the other studies contradicted those of Sanchez et al. [90], which sparked the controversy about using erythrocytes and leukocytes as crucial ingredients in the treatment of injured tendons, which were published by Arriaza et al. [91]. Most studies on white blood cells indicated that they could exert pro-inflammatory and catabolic effects on tenocytes [67,68,118,126,127].

The results of the clinical trials on the use of PRP in tendinopathies were mixed. Some trials indicated that LR-PRP injections resulted in better results than those given to patients with corticoid or saline [91,115,128,129]. On the other hand, some studies on whole blood and PRP did not show any beneficial effects [83,91,100,110,113,114,130,131,132].

The varying factors that affect the results of the clinical trials are also partly responsible for the mixed results. For instance, the number of injections, the type of tendon involved, and the patient’s age were all analyzed [133].

Recent studies on the development of stromal fibroblasts and human supraspinatus tendons from patients with ruptured ligaments revealed that these cells exhibited complex inflammation signatures [134,135]. These findings suggest that using PRP to treat these conditions could be a potential therapeutic option.

However, the lack of improvement in the functional and clinical outcomes of LR-PRP compared to the placebo or saline in patients with tendinopathies or ruptured tendons has raised doubts about its potential use in these conditions [136,137].

LR-PRP could also be beneficial for the healing of tendinopathies as it can stimulate the production of pro-inflammatory cytokines and catabolic substances on stromal cells [83,110,113,134,135,137]. This finding suggests that the effects of leukocytes on these cells could be partially derived from their pro-inflammatory and catabolic effects [119,137,138,139]. In addition, activating pro-inflammatory cytokines by injected leukocytes could potentially contribute to developing a non-resolving inflammation [83,133].

Studies on the development of osteoarthritis stem cells, stromal fibroblasts, and tendon stem cells from patients with tendinopathies revealed that the supernatant of LR-PRP could stimulate the production of pro-inflammatory cytokines. Compared to the supernatant of LP-PRP [54,119,133], the release of pro-inflammatory cytokines by the cultured cells was higher. This finding suggests that the use of this drug could be beneficial for the healing of these conditions [115].

In addition, Lipoxin A4, from platelets produced by arachidonic acid, has been shown to suppress the inflammatory processes in the tissues of people with tendinopathies and ruptured ligaments [136].

The lack of consistency in the results of the studies regarding the effectiveness of the PRP treatment process and the various preparation methods used for it has hindered the field’s advancement [93,96,100,103,104]. These elements can lead to misleading conclusions and prevent the public from being informed about its therapeutic potential. Studies on the use of PRP application in the foot and ankle for Achilles tendon pathology have been summarized in Table 3, with clear details.

### 3.5. Efficacy of PRP in Cartilage and Osteochondral Lesions of the Talus

Due to advances in the imaging technology, osteochondral lesions of the talus (OLT) are being increasingly recognized as a source of ankle discomfort. The conservative treatment is usually successful for small lesions, but surgical treatment is required for more extensive lesions or lesions that do not respond to the conservative treatment. Surgical procedures can be classified as either reparative or reconstructive.

PRP effectively treats osteochondral lesions (OLTs) and cartilage fractures in the talus. In a preclinical animal model, the treatment of OLTs with PRP resulted in improved histological scores and increased cartilage-like hyaline formation [123]. Clinical studies have shown that using PRP as an adjunct to the microfracture of OLT results in better outcomes than surgical repair [141,142].

In a randomized prospective trial study, Gurney et al. compared a total of 35 patients; patients in the control group (*n* = 16) received the microfracture surgery alone, whereas patients in the PRP group (*n* = 19) additionally received PRP therapy [142]. The authors found that after a mean follow-up of 16.2 months (range: 12 to 24 months), both groups showed significant improvements in clinical outcomes based on AOFAS scores, foot and ankle ability measures (FAAM), and VAS, although the PRP group outperformed the microfracture-only group. Patients with lesions larger than 20 mm in diameter were excluded from this investigation. The authors concluded that even if the addition of PRP to arthroscopic microfracture surgery for treating osteochondral lesions of the talus had shown better functional score status in the medium term, further research was required to evaluate the long-term.

In previous trials, most individuals with OCD lesions less than 15 mm in diameter were effectively treated with arthroscopic microfractures [143,144].

Mei-Dan et al. studied clinical and functional outcomes after three intra-articular PRP or HA injections [145]. At 28 weeks, the authors found that the PRP treatment greatly improved pain and function compared to HA. The data on PRP for osteochondral lesions of the talus are encouraging, but further studies are needed to evaluate the preparation, procedures, safety, and long-term outcomes before conclusions can be drawn.

A review conducted by Smyth et al. revealed that PRP could increase the number of chondrocytes and stem cells, the deposition of proteoglycans, and collagen formation. It also inhibited the effects of local catabolic cytokines; the researchers noted that using PRP during autologous osteochondral graft therapy significantly improved the graft integration and decreased the degree of cartilage degeneration [146].

There are still many questions to be resolved before the use of PRP can be considered an effective treatment for OLTs. For instance, the optimal combination of the PRP components should be studied. Those who were treated with allograft and PRP had similar results when it came to managing calcaneal fractures. They also exhibited better radiographic parameters and scores than those treated with the autograft alone [147]. The recommendations regarding using biologics to treat OLT can help clinicians make informed decisions regarding this difficult condition.

### 3.6. Efficacy of PRP Injections in Bone Healing

Bone regeneration with platelet-rich plasma (PRP) is designed to stimulate a healing response at fracture and fusion sites around the foot and ankle using platelet-derived products. In several preclinical studies, PRP has been shown to improve osteogenesis [8,21,22,23,24,25,26,27,28,29,30,31,44,45,46,47,48,49,50,51,52,53,54,55,56,57,58,59,60,61,62,63,64,65,66,67,68,69,70,71,72,73,74,75,76,77,78,79,80,81,82,83,84,85,86,87,88,89,90,91,92,93,94,95,96,97,98,99,100,101,102,103,104,104,105,106,107,108,109,110,111,112,113,114,115,116,117,118,119,120,121,122,123,124,125,126,127,128,129,130,131,132,133,134,135,136,137,138,139,141,142,143,144,145,146,147,148,149,150]. However, the translation of preclinical findings to in vivo bone repair has shown mixed results. Wei et al. conducted one of the few current studies on PRP and bone healing in the foot and ankle literature [147]. In a surgically-controlled, displaced intra-articular heel fracture, the authors examined the use of autografts against allografts with and without PRP. The AOFAS scores and imaging characteristics of the PRP and autograft groups were comparable at the long-term follow-up. Both were superior to the allograft group alone. Despite the high healing rate of heel fractures in the past, surgical treatment poses a significant risk of wound complications. This study did not provide enough information about wound healing with or without PRP to draw any conclusions.

Bibbo et al. investigated autologous platelet concentrate (APC) in elective surgery patients undergoing high-risk foot and ankle surgery [69]. Diabetic patients with neuropathy, immune, or nutritional compromise, a history of bone nonunion or delayed healing, prior surgery at the anticipated surgical site, or a history of open treatment after high-energy trauma were considered high-risk. Sixty-two high-risk patients were monitored with biweekly radiographs over six months to determine if they had radiographic healing. Patients who received APC alone had 40 days until healing, while those who received APC and autograft had 45 days until healing. The authors conclude that APC is a useful adjunct for high-risk patients those undergoing elective foot and ankle surgery to promote bone healing. The potential benefits of PRP in bone healing are intriguing, but further research is needed before any definitive conclusions can be drawn regarding using PRP in human bone regeneration.

### 3.7. Efficacy of PRP in Total Ankle Arthroplasty

Currently, numerous papers discuss the use of PRP in joint disorders, but few studies discuss the application of PRP in total ankle arthroplasty (TAR), and the topic is still debatable.

Barrow et al. [151] studied 20 patients with TAR who received PRP-assisted bone grafts for joint fusion. PRP was sprayed on the bone and prosthetic surfaces, mixed with the graft, and filled into the joint. This study showed 85% fusion within two months, 95% fusion within three months, and 100% fusion within six months, compared to the previous average of 62–82%. In a similar study, Coetzee et al. investigated the efficacy of platelet-rich plasma (PRP) in facilitating syndesmosis union after total ankle arthroplasty [70]. The retrospective analysis compared the outcomes of 66 patients who had PRP augmentation to those of 114 patients who did not. Eight weeks into the study, 61% of the control group had fused, and by the end of the study, 86% had fused. The fusion rate in the PRP group was 76% at eight weeks and 97% at six months. Patients with a cigarette use history showed a slightly increased fusion rate after PRP treatment. The authors conclude that carefully considering the patient history and risk factors (Table 2) is required before using PRP for fusion in ankle arthroplasty. More research is needed before further conclusions are drawn.

### 3.8. Efficacy of PRP in Ankle Osteoarthritis

Compared to hip or knee osteoarthritis, ankle osteoarthritis is quite rare [152,153]. In patients with knee OA, intra-articular PRP injections have enhanced clinical and functional outcomes [154,155,156,157,158]. A few articles on the use of PRP in ankle OA have been published. Repetto et al. studied 20 patients with symptomatic OA [156]. After a mean follow-up of 17.7 months, the authors found significant improvements in pain, function, and patient satisfaction.

In two studies, the effects of a combination of platelet-rich plasma and hyaluronic acid on the pain and function of patients with osteoarthritis were compared by Mei-dan et al. [145]. The study was conducted on 30 individuals with osteochondral lesions of the ankle. After 28 weeks, the patients were evaluated using the VAS, AHFS, and AOAFAS scores. After 28 weeks, the researchers noted that the patients who received the combination of PRP and hyaluronic acid experienced significantly better function and less pain.

Fukawa et al. studied 20 ankle OA patients who received three PRP injections every two weeks [157]. Up to 24 weeks after treatment, the authors found a significant improvement in pain and function. The greatest pain reduction occurred at week 12, after which the pain began to return to baseline levels but improved considerably.

In the study by Angthong et al. [159], they noted the clinical improvement in the VAS score after 16 months. They did not see changes in the joint after five months of follow-up. The researchers performed ultrasound-guided or scoped procedures on the subjects.

In vitro studies on chondrocytes revealed that the PRP increased their proliferation rate and stimulated matrix production. It also maintained the marker expression [148].

The analgesic effect of the PRP could be used as a potential drug for treating osteoarthritis. It could also enhance the secretion of hyaluronic acid. A recent study revealed that using PRP for treating osteoarthritis could be safely and effectively conducted with just a single injection [149].

Given the lack of available studies on the effectiveness of PRP at ankle OA, no definitive conclusions can be drawn. Limited data suggest a short- to medium-term benefit, but this must be compared with other injectable substances (corticosteroids, HA) in a long-term randomized controlled trial.

### 3.9. Efficacy of PRP Injections in Ankle Fractures

For over seven years, Wei et al. investigated all displaced type II heel fractures in their department [147]. A total of 276 fractures were randomly assigned to one of three treatment groups: autograft alone, allograft alone, or allograft with the addition of PRP. After one year, all fractures had completely healed, although there were no significant differences between the groups. At two and three years after the surgery, autografts alone and PRP-enhanced allografts were much less problematic than allografts alone and had much better radiographic outcomes (as measured by the Bohler angle, Gissane angle, and heel body dimensions). Clinical outcome assessments showed no differences between the groups in the degree of residual discomfort, walking ability, range of motion below the talus, or ankle-to-hindfoot alignment [147].

There is currently no evidence supporting the use of PRP in treating ankle fractures. However, limited data suggest that it cannot benefit ankle fracture recovery. It is important to conduct studies on the topic.

### 3.10. Efficacy of PRP Injections in Plantar Fasciitis

Plantar fasciitis has been the subject of many recent studies attempting to determine the function of PRP in its treatment [160,161]. Plantar fasciitis is the most common cause of heel discomfort in adults. The main cause is repeated microtrauma of the plantar fascia originating from the heel bone, leading to inflammation and degeneration. Some treatment possibilities include orthotics, splints, stretching exercises, physical therapy, extracorporeal shock wave therapy (ESWT), medications, injectables, and surgical release. Non-surgical treatments still fail in 10–15% of patients, resulting in persistent plantar fasciitis. Corticosteroids, autologous blood injections, and ESWT have all been tried with mixed results and risks, such as plantar fascia rupture after corticosteroid injection [162]. PRP is fascinating as a non-invasive method to improve plantar fascia recovery.

Martinell et al. treated 14 patients with chronic plantar fasciitis with PRP injections at three different puncture sites [163]. In total, 9 of the 12 patients showed significant improvement and reduced pain levels. They concluded that PRP is safe and effective at treating this disease. However, this study was limited by the lack of control treatment groups, such as the plantar stretching group. Similarly, Rabag et al. treated 25 patients with persistent plantar fasciitis with PRP injections and reported little or no functional limitation and significantly less discomfort in 23 of the 25 patients [164]. After the PRP injection, ultrasonography showed a significant increase in fascial thickness and signal intensity.

Aksahin et al. compared the PRP treatment with corticosteroids for persistent plantar fasciitis [165]. Thirty patients were injected with methylprednisolone and proparacaine, while the remaining 30 received PRP after the proparacaine injection. Pain ratings decreased sharply from 6.2 to 3.4 and 7.33 to 3.93 in the steroid and PRP groups. PRP appears to be a safer option when considering the risks of the corticosteroid treatment, such as sudden rupture, and the Carafino et al. study [166].

In patients with persistent plantar fasciitis, Kumar Jain et al. compared a single PRP injection with a corticosteroid injection [167,168]. They found that PRP and corticosteroids significantly improved VAS scores, modified Roles and Maudsley scores, the Foot and Ankle Outcome Instrument Core Scale, and the AOFAS Ankle-Hindfoot Scale, although there was no significant difference between the two groups.

According to Acosta-Olivo et al.’s statement, PRP showed the same effect as corticosteroids [169]. Aksahin et al. found significant improvement in VAS, modified Roles, and Maudsley scores with PRP and corticosteroid injections, but there was no significant difference between the two groups. Recent trials comparing PRP with corticosteroid injections and extended follow-up periods have found that PRP may have a more durable benefit than corticosteroids [170,171]. Monto found that PRP injections improved AOFAS scores after three months and that these effects lasted 24 months [171].

These findings contrast with those of corticosteroids, which improved AOFAS scores in the first three months but decreased to baseline at 24 months. Jain et al., in their investigation of correction functions and Maudsley scores, VAS scores, and AOFAS scores, observed similar effects [172]. PRP and corticosteroids scored the same at 3 and 6 months; however, PRP was much better at 12 months.

Singh et al. [173] combined the results of these and other studies in a systematic evaluation and meta-analysis. The authors concluded that PRP exceeded corticosteroids in VAS and AOFAS scores at three months but showed no difference in pain or function at 1, 6, or 12 months of the follow-up [173].

Based on the current evidence, it is still being determined whether the modest benefits claimed for PRP for persistent plantar fasciitis are sufficient to justify its efficacy. Most of the research on the topic shows that PRP injection significantly improves plantar fasciitis (Table 4). Further well-designed, prospective randomized controlled studies are needed to standardize the PRP injection in plantar fasciitis.

## 4. Efficacy of PRP Injections in Wound Healing and Diabetes-Related Issue

The diabetic foot is caused by peripheral neuropathy in diabetic individuals, leading to macroangiopathy and microangiopathy, arterial hypoperfusion, ulceration, and gangrene, which are the main consequences of diabetes mellitus [180].

The main factors that can prevent the successful healing of diabetic foot ulcer (DFU) are infection, poor tissue repair function, and the loss of growth factor secretion [181,182]. Based on the 22 clinical guidelines available, the current standard of care for treating diabetic foot involves a combination of pressure, shoes, adjunctive therapy, vascular assessment, wound off-loading, infection and glycemic control [8,9,10], and, most importantly, various surgical debridement techniques, such as minimally invasive metatarsal osteotomies applied to treat plantar diabetic foot ulcers [12].

One of the most important factors that can prevent the development and maintenance of DFU ulcers is promoting the healing process [183]. Although it is not as effective as these procedures, PRP can stimulate the healing process and prevent the development of DFU ulcers.

The study, which was conducted by Marx et al., provided valuable insight into the use of PRP in treating bone defects [6]. It has since been shown that treating these conditions can promote the wound-healing process and prevent the development of bacterial infections. In addition, the researchers noted that the PRP could help in the antibacterial treatment [184,185,186,187].

Mehrannia et al. [188] described a 71-year-old man with diabetes mellitus for 30 years who developed a large ulcer on his left foot due to diabetic neuropathy. Repeated infections made the plantar ulcer difficult to heal, putting the patient at risk of amputation. One injection of autologous PRP was administered to the foot after debridement, and he was discharged four days later. The plantar ulcer had healed after eight months of examination.

Karimi et al. [189] conducted a study. In a randomized controlled trial, 50 patients with diabetic foot ulcers were randomly assigned to two groups. After surgical debridement, the experimental and control groups were assessed for ulcer depth and surface area. The control group was covered with sterile dressings after frequent dressing changes, while the experimental group was coated with PRP gel dressing. After three weeks, the depth and surface area of the diabetic foot ulcers were significantly reduced in the experimental group compared to the control group, indicating that PRP gel promotes wound healing in diabetic foot ulcers.

Abdelhafez et al. randomly divided patients with a grade one diabetic foot into two groups. [190] The experimental group received PRP injections one or two times. Platelet gel was also applied to the ulcer; the control group used only platelet gel skin adhesives. Twenty-four patients (96%) in the experimental group healed completely after a 10-week treatment period. In comparison, 22 patients (88%) in the control group were completely healed by the end of the 10-week treatment period. When the 7 cm^2^ ulcers in the test group healed, it took significantly less time (2.6–16.0 days) than in the control group (3.4–27.0 days). The experimental group healed after 45 treatments and the control group after 54 treatments, indicating that PRP injection, combined with topical platelet gel, improved the healing rate of diabetic foot ulcers and shortened the treatment time.

Mohammadi et al. [191] published a study including 70 patients with diabetic foot ulcers. After local debridement, the ulcer area was estimated, and PRP gel (2 mL/cm^2^) was applied to the local ulcer. The results found that the mean healing time of the ulcers was 8.7 weeks after four weeks of treatment. The average reduction in the ulcer size was 51.9%. To explain how PRP can be used as a treatment for diabetes, the conservative treatment of foot ulcers can avoid diabetic consequences (e.g., amputation).

Using PRP can stimulate the production of various defense factors that can help the body fend off harmful bacteria [192]. These factors are beneficial in the treatment of DFU ulcers [193]. In addition, it can reduce the incidence of infections and the cost of treating DFU [194]. The autologous PRP can be used on patients with chronic wounds refractory to therapy. In previous studies, it has been shown that the treatment of diabetic foot and acute necrotizing fasciitis can benefit these patients [195,196].

Studies have also shown that au-PRP can help improve the healing process of chronic wounds, especially those related to DFU [197]. However, it is not always possible to use PRP for all diabetic patients [198]. For instance, some patients with severe chronic diseases, such as diabetes, anemia, and neuropathy, cannot benefit from the treatment of au-PRP. Alternatively, al-PRP has also been shown to promote the healing process in chronic wounds [199,200]. Although PRP can be safely used for autologous procedures, it is not always possible to harvest it from diabetic patients due to their condition. Poor cell activity or a low number of platelets could also prevent the use of the PRP.

According to He et al. [201], in 2020, the increasing number of healthy and younger individuals donating their allogeneic PRP led to the development of new strategies for treating chronic wounds. Al-PRP could potentially be an effective alternative to au-PRP at treating chronic wounds.

Different types of PRP have various efficacies on wound healing; there has been controversy on the subject; studies have shown that PRP, according to their composition, can accelerate wound healing (Table 5); al-PRP and au-PRP should be studied and considered more by the physician treating diabetic foot ulcer and wound healing.

## 5. Summary

### What Needs to Be Remembered

PRP utilization in the foot and ankle field is increasing. It is currently utilized to treat various foot and ankle conditions, such as fracture, osteoarticular joint fusion, osteoarthritis, Achilles tendon disease, cartilage and osteochondral lesions of the talus, plantar fasciitis, diabetic foot, etc. However, we must be aware that there are no clear indications for PRP therapy at this stage. It is unknown whether this is a non-surgical treatment or an adjuvant to surgery. Regarding the preparation process of PRP, including blood separation and related characteristics, there is a lack of uniform standards in the commercial equipment set offered by domestic and international enterprises for preparing PRP. The ideal platelet concentration in PRP after isolation is yet unknown. According to studies, the link between platelets, the concentration of growth factors in platelets, and their reparative effects on wounded tissues may not be linear.

At the location of the damage, cell surface receptor sites that bind to growth factors, once the amount of growth factors exceeds the number of appropriate cell surface receptors, and excess free growth factors in tissues may block cell activity, hence reducing the clinical therapeutic impact of PRP [45]. In addition, past investigations have demonstrated that normal individuals with equal platelet counts have varying growth factor concentrations. This discrepancy may impact the clinical therapeutic efficacy of PRP for several disorders, impacting the precision of data analysis during the study process. After isolation, leukocytes and monocytes may exist in PRP. Their function in the initial phases of the inflammatory response is well understood. According to some researchers, the white blood cells in PRP can eliminate necrotic tissue and dangerous germs. However, other studies believe that the protease and oxygen-free radical mediated by leukocytes may hinder tissue repair when PRP is employed in an inflammatory response. With the rise in leukocyte content, the expression of several genes involved in tissue catabolism was considerably increased in the wounded tissue. According to the current report, it is uncertain whether PRP separation includes leukocytes. There are presently no broadly accepted therapy guidelines for the clinical application of PRP in foot and ankle surgery. There is no universal requirement for the dose of each PRP injection, the total number of injections, or the time interval between each injection, especially when PRP is used for the non-surgical therapy of specific disorders. In foot and ankle surgery patients with thrombocytopenia, hemodynamic instability, sepsis, and bone graft infection, PRP should be avoided. Using PRP in blood extraction and separation from these individuals may exacerbate hemorrhaging, shock, and the spread of infection.

Patients with bone tumors should also be used with caution while receiving PRP. Multiple growth factors present in PRP have the potential to promote tumor growth. In addition, the intra-articular injection of a high concentration of PRP may induce joint discomfort and transitory joint dysfunction; therefore, it is essential to communicate with patients and gain their trust and cooperation before administering PRP therapy. The clinical application of PRP in the foot and ankle field recently has shown promising prospects. The majority of patients were pleased with PRP’s therapeutic effect. However, the number of published clinical studies is still very limited, and many of these works of literature are empirical retrospective reports. In the future, rigorous and high-quality randomized controlled studies with large samples are still needed to further confirm PRP’s application value and develop targeted standards for PRP autologous blood separation and clinical application guidelines for foot and ankle fields to serve patients better.

In this review, the most recent studies were summarized, analyzed, and solidly discussed like never before, with supplementary tables and diagrams of the yearly publications of PRP in the foot and ankle field, making our study unique and full of useful knowledge for orthopedic physicians. The results of these studies will pave the way for developing effective and efficient procedures for treating various musculoskeletal conditions in the foot and ankle in the future. 

## Figures and Tables

**Figure 1 jcm-12-01002-f001:**
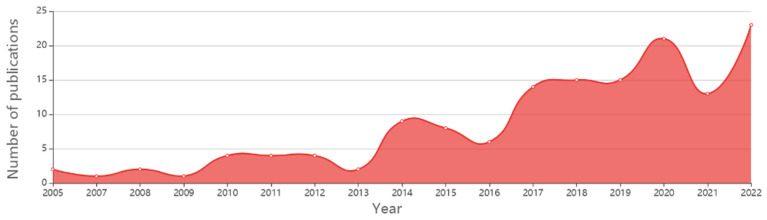
A diagram of worldwide publications on PRP use in foot and ankle, giving the number of articles per year.

**Figure 2 jcm-12-01002-f002:**
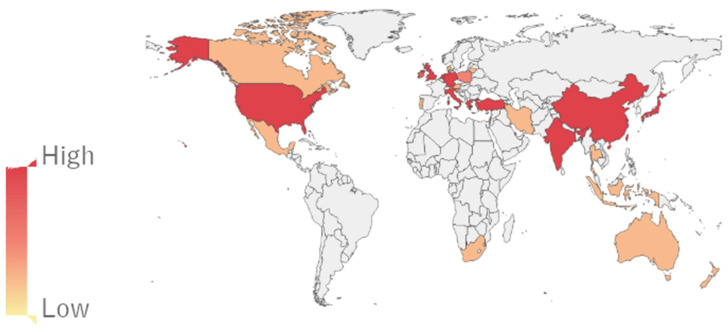
Worldwide research status of PRP in foot and ankle fields.

**Table 1 jcm-12-01002-t001:** Effect of growth factors present in PRP.

Name	Abbreviation	Cell Source	Functions	References
Epidermal growth factor	EGF	Platelets, macrophages, epithelial cells, eosinophils	Proliferation and differentiation of epithelial cells	Ren, Xiaochen, et al., 2020 [32]
Transforming growth factor- beta	TGF-β	Platelets, macrophages, osteoblasts, immune chondrocytes, T lymphocytes	Fibroblast proliferation, collagen synthesis, bone matrix formation, inhibition of bone resorption	Elder and Thomason, 2014 [33]
Insulin-like growth factor	IFG	Plasma, epithelial, and endothelial cells, fibroblasts cells, smooth muscle, osteoblasts, bone matrix	Fibroblast chemotaxis, proliferation and osteoblast differentiation, bone matrix formation, the growth and repair of skeletal muscle	Creaney and Hamilton, 2008 [34] Martínez et al., 2016 [35] Kleplová et al., 2014 [36]
Vascular endothelial growth factor	VEGF	Basophils	Angiogenesis of endothelial cells, migration, and mitotic cells, chemotaxis of macrophages and granulocytes, the vasodilation	Bai et al., 2014 [37] Yamakawa et al., 2019 [38]
Platelet derived growth factor	PDGF	Platelets, macrophages, smooth muscle cells, bone matrix, epithelial cells, endothelial cells,	Mitogenesis, angiogenesis, regulation of function of other cells and growth factors (stimulation of fibroblasts and osteoblasts, induction of cell differentiation, catalyzing the effects of other growth factors on other cells macrophages)	Martínez et al., 2016 [35] Kleplová et al., 2014 [36]
Connective Tissue Growth Factor	CTGF	Platelets through endocytosis from extracellular environment in bone marrow	Promotes angiogenesis, cartilage regeneration, fibrosis, and platelet adhesion	Nikolidakis et al., 2008 [39] Chen, Zihao, et al., 2020 [40]
basic Fibroblast Growth Factor	bFGF	bone marrow stem cells, macrophages	Stimulate bone marrow stem cells’ differentiation into bone; indicate severe bone lesions; induce calcium deposition; support bone marrow stem cells’ expansion	Kawaguchi et al., 2010 [41] Hata et al., 2013 [42] Bai et al., 2014 [37] Cheng et al., 2014 [43]

**Table 2 jcm-12-01002-t002:** Summary of risk factors for impaired bone healing.

Trauma
Smoking
Diabetes
Older age (+50)
Open injury
Corticosteroid use
Immunosuppression
Multiple (>2) surgeries
Peripheral vascular disease
History of infection or active infection
Nonunion or pseudoarthrosis at site

**Table 3 jcm-12-01002-t003:** Studies on the use of PRP application in foot and ankle for Achilles tendon pathology.

Authors and Years	PRP Class	Number of Injections	Follow-Up (Wks/Mos)	Achilles Tendon Lesion	Outcome	Sample Size		Level of Evidence
						PRP	Control	
Kearney et al., 2021 [103]	LR-PRP	Once/-/3	2 wks, 3 and 6 mos	C-AT (>6 mos)	There was no significant difference in VISA-A scores between the PRP group and the sham group at 6 months.	121	119	I
Thermann et al., 2020 [106]	N/R	Once/-/NR	6 wks, 3, 6, and 12 mos	C-AT (>6 mos)	There was no significant difference between the PRP and control group.	17	19	I
Keene et al., 2019 [113]	N/R	Once/-/4	1, 4, 7, and 24 wks	A-ATR (<12 days)	There was no significant difference in muscle-tendon function between the PRP and control group.	114	116	I
Liu et al., 2019 [97]	N/R	Once/-/4/Once/-/6	6, 12, and 24 weeks, 12 mos	C-AT	Significant differences in the VISA-A were not observed between the PRP and placebo groups after 12 weeks. VAS scores after 6 and 24 weeks were not significantly different. VAS scores and Tendon thickness were significantly different in the 12th week.	Meta-analysis of 5 RCTs (*n* = 189)		I
Boesen et al., 2017 [96]	LR-PRP	4 times/2-wks/4	6, 12, and 24 wks	C-AT (>3 mos)	VISA-A, VAS, and Tendon thickness improved in all groups at 6,12, and 24 wks (*p* < 0.05).	20	20	I
Krogh et al., 2016 [100]	LR-PRP	Once/-/6	3, 6, and 12 mos	C-AT (mean 33 mos)	There was no significant difference between the PRP and control group VISA-A, VAS scores, and Tendon thickness at 3 mos.	12	12	I
Kirschner et al., 2021 [107]	LR-PRP	1	6, 52, and 104 wks	AT	There were no significant differences between the LR-PRP and control group groups at 6, 52, and 104 wks (*p* > 0.05).	21	19	II
Boesen et al., 2020 [140]	N/R	Once/-/4	8 wks, 3, 6, and 12 mos	A-ATR (<3 days)	There was no significant difference between the PRP and control group at 12 mos.	19	19	II
Abate et al., 2019 [93]	N/R	3(1/w/3 W)	3 and 6 mos	N-ATR	There was no significant difference between the PRP and control group in pain and function at 3 and 6 mos.	46	38	III
Erroi et al., 2017 [92]	N/R	Once/-/3	2, 4, and 6 mos	I-AT	There was a significant difference between the PRP and control group VISA-A and VAS scores at 2, 4, and 6 mos.	21	24	III
Hanisch et al., 2019 [98]	LR-PRP/ LP-PRP	LR-PRP: Once/-/5–6 LP-PRP: Once/-/5	2, 8, and 48 wks	C-AT (>6 mos)	There was a significant difference between the PRP and control group VISA-A and VAS scores between the LP-PRP and LR-PRP group.	36 (LR-PRP)	48 (LR-PRP)	IV
Zou et al., 2016 [125]	N/R	Once/-/NR	3 wks, 3, 6, 12, and 24 mos	A-ATR (<3 wks)	The PRP group had an improved ankle range of motion compared with the control group at 24 months.	16	20	N/R
De Carli et al., 2015 [120]	N/R	2 times/2-wks/4	1, 3, 6, and 24 mos	A-ATR	There was no significant difference between the PRP and control group VISA-A and VAS score at 1, 3, 6, and 24 mos.	15	15	IV
Schepull et al., 2011 [102]	AU-PRP	Once/-/10/4	7, 19, and 52 wks	A-ATR (<3 days)	There was no significant difference in heel raise index and in elasticity modulus between the PRP and control group at 7, 19, and 52 wks. The Achilles Tendon Total Rupture Score in the PRP group at 7, 19, and 52 wks was lower.	15	14	II

LR-PRP: leucocytes rich PRP; AU-PRP: autologous PRP; N-AT: non-insertional Achilles tendinopathy; I-AT: insertional Achilles tendinopathy; CAT: chronic Achilles tendinopathies PRP: platelet-rich plasma; C-AT: chronic Achilles tendinopathy; wks: weeks; mos: months; PRP acquiring ratio: blood volume (mL): PRP acquiring ratio (mL); N/R: not reported.

**Table 4 jcm-12-01002-t004:** Studies on the use of PRP application in foot and ankle for Plantar fasciitis.

Authors and Years	PRP Class	Number of Injections	Follow-Up (Wks/Mos)	Outcome	Sample Size		Level of Evidence
					PRP	Control	
Vetrano et al., 2013 [79]	not reported	Once/2 wks/2	2, 6, and 12 mos	The PRP group showed a significant difference in improvement than the ESWT group in VISA-P and VAS scores at 6 and 12 mos.	23	23	I
Tiwari et al., 2013 [174]	leukocyte-rich PRP	2–3/1 wks/5	1, 3, and 6 mos	A significant improvement in the VAS score was observed between the PRP and placebo groups after 3 and 6 mos.	30	30	I
Monto et al., 2014 [171]	leukocyte-rich PRP	Once/-/3	3, 6, 12, and 24 mos	PRP was more successful and long-lasting than cortisone injection in treating chronic plantar fasciitis.	20	20	I
Jain et al., 2015 [172]	leukocyte-rich PRP	Once/-/2	1, 3, 6, and 12 mos	There was no significant difference between the PRP and control group in plantar fascia thickness at 1, 3, 6, and 12 mos.	30	30	I
Sherpy et al., 2016 [175]	leukocyte-rich PRP	Once/-/1	1, 5, and 3 mos	There was no significant difference between the PRP and steroid groups at 3 months.	25	25	I
Haghighat et al., 2016 [170]	leukocyte-rich PRP	Once/-/1	1, 3, and 6 mos	A significant improvement in pain severity and physical limitation was observed between the PRP compared to the placebo groups at 3 mos.	16	16	I
Mahindra et al., 2016 [176]	not reported	Once/-/1	3 wks, 3 mos	Both PRP and control group were significant in treating plantar fascia thickness at 3 mos.	25	25	I
Acosta-olivo et al., 2017 [169]	not reported	Once/-/3	2, 4, 8, 12, and 16 weeks	There was no significant difference between the PRP and control group in pain and function at 2, 4, 8, 12, and 16 weeks.	14	14	I
Shetty et al., 2018 [177]	not reported	Once/-/1	18 mos	PRP significantly improved pain, function, and general health compared to the corticoid group at 18 mos.	30	30	I
Peerbooms et al., 2019 [148]	not reported	Once/-/1	4, 12, and 24 wks, 12 mos	The PRP group showed significantly lower Foot Function Index Disability scores than the control group at 12 mos.	46	36	I
Huang et al., 2020 [168]	not reported	Once/-/2	1, 3, and 6 mos	There was a statistically significant better long-term functional improvement in PRP than in the control group in treating plantar fasciitis.	295	293	I
Hurley et al., 2020 [178]	not reported	Once/-/2.5/5	1, 1.5, 3, 6, and 12 mos	The PRP group showed better results than the placebo at 6 and 12 mos.	239	240	I
Hohmann et al., 2021 [179]	not reported	Once/-/3	1, 3, 6, 12, and 18 mos	The PRP group had a better VAS score than the control group at 6 and 12 mos.	457	354	I

**Table 5 jcm-12-01002-t005:** Studies on the use of PRP application in foot and ankle for diabetic foot ulcer and wound healing.

Authors, and Years	Source	Number Centrifuge Time	Frequency	Wound Duration per Week		Preparation	Follow-Up per Week	Outcome	Type of Study
				PRP	Control				
Jeong et al., (2010) [202]	AL	1	2/W	12.4	10.1	Blood bank	NR	The PRP group had 79% complete wound healing compared with 46% in the control group (*p* < 0.05). In the PRP–treated and control groups, full healing took 3 to 12 weeks and 6 to 12 weeks, respectively (*p* < 0.05). PRP-treated and control groups had wound shrinking of 96.3% and 81.6%, respectively (*p* < 0.05). No adverse events were reported.	RCT
Liao et al., (2020) [203]	AL	2	2	60		Homemade	NR	After 30 days, the AL-PRP group had less inflammatory exudation than the control. AL-PRP-treated chronic wounds healed faster than controls (first week: t ¼ 7.6349, *p* < 0.05; third week: t ¼ 18.456, *p* < 0.05). No rejections occurred.	RCT
He et al., (2020) [201]	AL	2	2/w	AL-PRP (*n* = 20) AU-PRP (*n* = 25)	30	Blood bank	12	The wound healing times of the AL-PRP group and AU-PRP group were significantly shorter than those of the control group.	OS
Saldalamacchia et al., (2004) [204]	AU	2	1/w	NR	NR	Blood bank	5	The platelet gel group had 71% complete healing and the conventional treatment group had 29% (OR 6.2; 95% CI 0.6–63). No adverse effects were reported.	OS
Saad Setta et al., (2011) [205]	AU	2	2/w	NR	NR	Homemade	NR	Time to complete wound healing was faster in the PRP group (*p* < 0.005). PRP speeds chronic diabetic foot ulcer healing.	RCT
Li et al., (2015) [206]	AU	2	2/w	4.28	3.28	Homemade	163	Proportion of complete-healed diabetic foot ulcers was high than the control. No wound complications occurred. No recurrences.	RCT
Karimi et al., (2016) [189]	AU	1	1/w	NR	NR	Homemade	3	Wound size was significantly greater in the PRP group compared to the control group (*p* = 0.019). Wound size was significantly reduced in both PRP and control groups (*p* < 0.05).	RCT
Ahmed et al., (2017) [207]	AU	2	2/w	12.5	11.5	Homemade	12	The healing rate was 86% in the PRP group versus 68% in the control group. The PRP group has lower rate of wound infection.	OS
Driver et al., (2006) [208]	AU	1	2/w	NR	NR	Kit	24	Wound healings was faster in the PRP group compared with the control. No wound complications. No adverse events reported.	RCT
Kakagia et al., (2007) [209]	AU	1	1/w	20	19	Kit	8	The proportion of complete-healed diabetic foot ulcers reached statistical significance. There was a significantly greater reduction in all three groups of ulcers (all *p* < 0.001).	RCT

NR: Not reported; RCT: Randomized controlled trial; OS: Observational study; OCS: Observational cohort study; AL: Allogenic. AU: Autologous; W: Week.

## Data Availability

Not applicable.

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
