# Peer review of "Advances in the Clinical Application of Platelet-Rich Plasma in the Foot and Ankle: A Review"

_jcm, 2023, doi:10.3390/jcm12031002_

Round 1

Reviewer 1 Report

In general, the review is cumbersome to read. The introduction needs to be rewritten and improved also in light of the other subsections of the manuscript. At the end of the introduction, the aim should be clearly reported as well as the outline of the review. Authors should avoid simply listing works. The various sections must be linked by a minimum of introduction. Perhaps the foot section could be well separated from the ankle section and a brief introduction to the foot section and one to the ankle section could help the reader to follow the review better.

Specific comments

·      

·         I suggest to avoid the used of abbreviations in the title.

·         Affiliation’s number is missing.

·         Line 12: “PRP” should be defined. Abbreviation should be consistently used throughout the manuscript (for example line 13).

·         Line 26: Risk factors should be reported.

·         Lines 25-26: references should be added.

·         Line 37: attractive alternative compare to what? Surgery?

·         Line 41: “It is used to remove red blood cells”. This part should be better explained. It seems that PRP is used to remove blood cells, while red blood cells are removed to produce PRP.

·         Lines 45, 55: the full stop should be moved after the citations.

·         Lines 45-47: “In the presence of trauma, activation of platelets results in degranulation and release of the alpha particles. Some factors that can be released include the growth factors PDGF, IGF, and TGF-1”. References should be added. Abbreviations should be defined. The authors should check all the manuscript and define abbreviations at first mention. I suggest to delete

·         Line 47: TGF-1 is TGF-Beta 1?

·         Lines 48-51: this part should be improved. Foot and ankle disorders should be added. Information about epidemiology/incidence of these disorders should be added etc.

·         Lines 53-54: this sentence should be rewritten. Did the authors mean that injury is one of the most common complications during foot and ankle arthroscopic surgery?

·         Figure 1: y axis should be defined instead of reporting the number of papers for each year in the figure’s caption.

·         Figure 2: colors should be defined.

·         Table 1: references for each growth factor reported should be added.

·         Line 62: references should be added.

·         Lines 69-70: here the authors report that further evidence is needed to support the effectiveness of PRP in treating foot and ankle disorders jest before the aim of the review. The reader is a little disconcerted to read this sentence even before the purpose. If there is not enough evidence, what is the point of writing and reading this review? This sentence should be included in the conclusions.

·         The subsections should be numbered following the guidelines of the journal.

·         Lines 91-92; 183-184: references should be added.

·         Lines 115-125: why did the authors report this part here? Is there only one published review on PRP and foot/ankle conditions? This part should be moved into the introduction to explain the gap in the literature and the need to write the current review.

·         Before starting with subsections like “Effectiveness of PRP use in Ankle fusion” etc., the authors should add a paragraph of introduction to these topics. Also, authors should try to avoid simply listing works.

·         Line 130: It seems that PRP was not injected. Is that correct?

·         Line 132: it is unclear in which tissue or where TGF-beta and PDGF were evaluated.

·         Lines 135-143: this study is confused. Different treatments and different foot/ankle conditions.

·         Section “Effectiveness of PRP use in Ankle fusion”: the authors reported only two studies. Are there other studies?

·         Table II is out of the scope of the review.

·         For each study reported, the authors should clearly write how PRP was produced and applied (timing, dose etc).

·         Line 174: The listed studies should be explained.

·         Line 175:  what is the standard methodology for the authors?

·         Lines 176-180: this is valid also for the other conditions.

·         Line 193: AT is unclear.

·         Table III should be improved reporting all the studies and the outcome should be better explained. Tables summarizing the studies on Achiles tendinopathy, in surgical augmentation in the AT and PRP Injections in Achilles tendon rupture should be added.

·         Lines 372-376, 489-491,491-493: references should be provided.

·         Lines 391-392: details should be added.

·         Line 418: “Curr Rev Musculoskelet Med (2018) 11:616–623” should be correct.

·         Table IV: outcomes should be better reported.

·         Lines 543-545: references should be added. Diabetic foot ulcers can be treated also by different surgical techniques. For example, minimally invasive metatarsal osteotomies are applied to treat plantar diabetic foot ulcers.  It should be added.

·         Summary should be better explained.

Author Response

Author's Responses to the Reviewer comments (Reviewer 1) 

Authors: We appreciate the Editor's and Reviewers' constructive remarks on our work, which have strengthened it. We responded to the criticisms shown on the attached pages, and the changed text in the manuscript is highlighted in red. We hope the revised work will be accepted for publication in the Journal of Clinical Medicine. 

1· I suggest to avoid the used of abbreviations in the Title

Response: thanks for your comment.

The abbreviation in the Title has been changed and written in red font in the manuscript as follows:

"platelet-rich plasma", 

the new Title is "Advances in the clinical application of platelet-rich plasma in the Foot and Ankle: A Review."

2· Affiliation's number is missing. 

Response: we are glad about this comment.

The affiliation number has been added; it is written in red font in the manuscript.

"1Department of Orthopaedics, Xiangya Hospital of Central South University, 410008 Changsha, Hunan, China"

3· Line 12: "PRP" should be defined. Abbreviation should 

be consistently used throughout the manuscript (for 

example line 13). 

Response: thanks for the comment.

Yes, we are sorry for this; as requested PRP abbreviation has been defined as follows and is written in red font in the manuscript.

"platelet-rich plasma(PRP)"

also, an abbreviations list was provided at the end of our work; based on that, we did not first provide it; we are sorry.

4. Line 26: Risk factors should be reported.

Response: We appreciate your comment.

Our team have suppressed this part of the manuscript as you suggested that the introduction should be rewritten. Thanks for that suggestion because it helps us to ameliorate our manuscript.

5· Lines 25-26: references should be added. 

Response: We appreciate your helpful comments.

We are sorry for the Mistake. As you suggested introduction was reformulated, so this part of the manuscript has been suppressed.

6· Line 37: attractive alternative compare to what? 

Surgery?

Response: we appreciate your impressive comment.

This part of the manuscript has been suppressed.

However, we answer the question to be clear.

Attractive alternative Compared to other orthobiologics substances that have been used in the field recently, we did not mean to compare PRP to surgery. We said PRP is easier to use than other orthobiologics defined below.

 "Orthobiologics" are substances that orthopaedic surgeons use to help injuries heal more quickly. 

Common orthobiologics include "platelet-rich plasma", bone marrow concentrate, certain fat grafts, and birth tissues.

7· Line 41: "It is used to remove red blood cells". This part should be better explained. It seems that PRP is used to remove blood cells, while red blood cells are removed to produce PRP. 

Response: we appreciate your comment.

"It is used to remove red blood cells" has been removed from the manuscripts because it is a vague phrase and because the PRP preparation method has been explained in detail below in the manuscript.

8· Lines 45, 55: the full stop should be moved after the 

citations. 

Response: we thank you for your comment.

As requested, the full stops have been moved after the citations.

8· Lines 45-47: "In the presence of trauma, activation of platelets results in degranulation and release of the alpha particles. Some factors that can be released include the growth factors PDGF, IGF, and TGF-1". 

References should be added. Abbreviations should be defined. The authors should check all the manuscript and define abbreviations at first mention. I suggest to delete 

Response: thank you for your comments; we appreciate it.

This part of the manuscript was suppressed, and the introduction has been reformulated.

 About the abbreviations, we are sorry again, as we provided an abbreviation list at the end of our work, we tough it was enough, but thanks to your comments, we resolved the ambiguity.

The abbreviations mistakes have been corrected; thanks.

9· Line 47: TGF-1 is TGF-Beta 1?

Response: thanks for your comments

we meant TGF-Beta, written in red in the manuscript as "Transforming growth factor beta."

10· Lines 48-51: this part should be improved. Foot and ankle disorders should be added. Information about epidemiology/incidence of these disorders should be added etc. 

Response: we appreciate your comments.

*As requested, the foot and ankle disorders have been added, and the modified part is written in red font in the manuscript as follows:

"foot and ankle surgery because of multiple foot and ankle disorders such as Achilles tendon diseases, Adult-acquired flatfoot deformity, Ankle fracture, Ankle sprain, Midfoot arthritis, Osteochondral defect of the talus, Plantar fasciitis, which can severely affect patients daily lives and is usually treated conservatively."

*Pieces of information about the epidemiology/incidence of these foot and ankle disorders have not been added to the present manuscript because we did not conduct a study on a specific group or ethnicity. After our team searched the topic, we noticed many pieces of information depending on the country, province, ethnicity, or the information was too old. Below are 3 articles on epidemiology/incidence in the foot and ankle to show that we have researched the topic respectfully.

Reed, Lloyd F., et al. "Prevalence and risk factors for foot and ankle musculoskeletal disorders experienced by nurses." BMC musculoskeletal disorders 15.1 (2014): 1-7.

https://doi.org/10.1186/1471-2474-15-196

Hansen, Regina, Naohiro Shibuya, and Daniel C. Jupiter. "An updated epidemiology of foot and ankle fractures in the United States: complications, mechanisms, and risk factors." The Journal of Foot and Ankle Surgery (2022).

https://doi.org/10.1053/j.jfas.2022.01.010

Lazzarini, Peter A., et al. "Prevalence of foot disease and risk factors in general inpatient populations: a systematic review and meta-analysis." BMJ open 5.11 (2015): e008544.

https://doi.org/10.1136/bmjopen-2015-008544

11· Lines 53-54: this sentence should be rewritten. Did the authors mean that injury is one of the most common complications during foot and ankle arthroscopic surgery? 

Response:  thanks for your comment.

Yes, the sentences have been rewritten, and we are sorry if, at first, you get confused. The modified sentence was written in red font in the manuscript as follows.

"Foot and ankle surgery has the highest complication rate and may be associated with articular cartilage injury, wound complications, instrument breakage, infection, nerve, tendon, and ligament injury, and long-term nerve damage [10,11]."

12· Figure 1: y axis should be defined instead of reporting the number of papers for each year in the figure's caption. 

Response: we appreciate your comment.

As requested, Y and X's axis have been defined and written on the figure directly," Y-Axis = the number of publications, and X axis = year."  

13· Figure 2: colors should be defined. 

Response: Thanks for your comment.

As you requested, we defined the colours; we did it directly on the figure.

"Red= high research zone, Brown= low research zone"

 14· Table 1: references for each growth factor reported 

should be added. 

Response: we appreciate your impressive comment.

The references for each growth factor have been added as requested.

Thirteen references in total; please have a look at table 1. The modified part is in red font.

15· Line 62: references should be added. 

Response: thanks for your comment.

As we modified the introduction of the present manuscript, as you suggested, this part has been cancelled.

16.· Lines 69-70: here, the authors report that further evidence is needed to support the effectiveness of PRP in treating foot and ankle disorders jest before the aim of the review. The reader is a little disconcerted to read this sentence even before the purpose. If there is not enough evidence, what is the point of writing and reading this review? This sentence should be included in the conclusions. 

Response: thanks for your wonderful and good comment.

As suggested, the phrase "There still needs to be more evidence supporting the effectiveness of PRP in treating various foot and ankle pathologies." has been removed as it can confuse our lectors and will be added when we reformulate our conclusion.

Moreover, our conclusion has been modified thanks to your good suggestion.

17· The subsections should be numbered following the 

guidelines of the journal. 

 Response: the subsections have been numbered following the journal's guidelines; thanks for your comment.

18· Lines 91-92; 183-184: references should be added. 

Response: thanks for your good remarque.

As requested, the references were added. 

Lines 91-92 [38] and 183-184 [60], written in red font in the present manuscript.

19· Lines 115-125: why did the authors report this part here? Is there only one published review on PRP and foot/ankle conditions? This part should be moved into the introduction to explain the gap in the literature and the need to write the current review. 

Response: We are happy about this comment which helps us to improve our work; thanks.

As requested, this part was removed and reformulated while rewriting the introduction. Written in red in the manuscripts.

20.Before starting with subsections like "Effectiveness of 

PRP use in Ankle fusion" etc., the authors should add a 

paragraph of introduction to these topics. Also, authors 

should try to avoid simply listing works. 

Response: thanks for your comment.

As requested, a paragraph has been added to introduce the topics and is written in red font in the manuscript.

*"PRP's influence on bone healing has been extensively studied in vitro and in vivo.[48-54]The hypothesis is that platelets and their growth factors will boost osteopontin, osteoprotegerin, osteoblast, osteoclast-like cells, and differentiation of myoblasts and osteoblastic cells. [52-56] PRP's effectiveness in bone healing is still debated. Many studies indicate promise, while others show little difference between PRP and control or standard products." 

*Most of our paragraphs have been introduced, and those that have not been introduced have been completed with an introduction.

*sorry if you feel that a part of the work has been listed, but maybe it is just because we try to divide paragraphs according to studies; and in our conception, yes, we try to make every part a subsection. Thanks for your comments; we will try our best to make it clearer and more understandable.

21· Line 130: It seems that PRP was not injected. Is that 

correct? 

Response: Thanks for your comments

our review team, after your comments, decided it was necessary to reformulate the whole paragraph and bring more information; we hope you will find it clear now.

*And to reply to your question, no PRP was not injected but combined with the autograph during the second surgery.

It is written in red font in the manuscript.

"PRP combined with autograft was applied to the nonunion in the second revision surgery."

22· Line 132: it is unclear in which tissue or where TGF

beta and PDGF were evaluated. 

Response: we appreciate your comment.

As mentioned above, our team has rewritten the whole paragraph. The modified part is in red font in the manuscript and answers directly to your interrogation; thanks.

"The authors also compared the growth factor concentrations in the hematoma at the fracture site in patients with nonunion and union and found that the concentrations of PDGF and TGF-β in nonunion hematoma were significantly lower than those in fresh fractures. "

23· Lines 135-143: this study is confusing. Different treatments and different foot/ankle conditions. 

Response: thanks for your helpful comment.

It is clear now that we have reformulated the whole paragraph, and sorry for previously making it hard to understand.

Another thing is that we reported what was in these articles. For now, one of the most difficult things in PRP therapy is treatment heterogeneity, which is why a consensus has not been reached yet.

24· Section "Effectiveness of PRP use in Ankle fusion": the 

authors reported only two studies. Are there other 

studies? 

Response: we are glad about your comment.

Yes, previously, we just reported two studies on the topic, but now that we have reformulated the whole paragraph, we have more studies; the difficulty, as mentioned previously, is that the researches on PRP use in the ankle field are few, especially in the clinical research, and another thing is that we divided studies into subsections. We hope you are satisfied with this paragraph written in red font in the manuscript.

25. Table II is out of the scope of the review. 

Response: thanks for your comment.

Table II is not independent, referring to risk factors that can slow bone healing in patients with foot and ankle disorders during PRP therapy. You can see we purposefully made the table to refer to it when it comes to enumerating the risk factors of nonunion in patients.

26.· For each study reported, the authors should clearly write how PRP was produced and applied (timing, dose etc.). 

Response: thanks for your comments.

Yes, we would like to provide all this data, but not all studies provide such data, making it difficult to conclude when it comes to PRP efficacy.

One of the biggest problems in PRP therapy is that there is no consensus on its production dosing, time of injection etc. Most studies just mentioned they use it during surgery, for example, or combined with a bone graft. Moreover, the data we got are all inserted in the present manuscript in different tables and figures. We continue doing our best to make it more readable to lectors.

27· Line 174: The listed studies should be explained. 

Response: we appreciate your comment.

All of the listed studies have been explained forward in the manuscript, so to prevent redundancy, we abstained from it. Thanks for your wonderful comment.

28· Line 175: what is the standard methodology for the 

authors? 

Response: thanks for your wonderful comment。

Sorry for this little Mistake that has been corrected. We meant "standard procedure."

29· Lines 176-180: this is valid also for the other conditions.

Response: thanks for your comment.

Yes, you are right; this is also valid for other conditions, but as it is also valid for Achilles tendinopathy, we try to be specific as we divide our work into subsections. Thanks for your good observation.

30· Line 193: AT is unclear.

Response: thanks for your wonderful observation.

This Mistake has been modified, and AT has been defined and written in red in our manuscript"Achilles tendinopathy."

31. Table III should be improved, reporting all the studies, and the outcome should be better explained. Tables summarizing the studies on Achilles tendinopathy in surgical augmentation in the AT and PRP Injections in Achilles tendon rupture should be added.

Response: thanks for your comment.

we have modified the outcomes in all the table, it was a hard work for our team to go bacnk there again and do all the search ,we also try our best to improve the table hope you appreciate our work,thank you.

32· Lines 372-376, 489-491,491-493: references should be 

provided. 

Response: Thanks for your wonderful comment.

the references were added as requested, respectively 

Lines 372-376" [56,57,108,116,117]."

Lines 489-491,491-493"[138]"

33· Lines 391-392: details should be added. 

Response: we appreciate your comment.

As requested, details of the study have been added as follows.

"In a randomized prospective trial study, Gurney et al. compared a total of 35 patients; patients in the control group (n = 16) got microfracture surgery alone, whereas patients in the PRP group (n = 19) additionally received PRP therapy[133]. The authors found that after a mean follow-up of 16.2 months (range: 12–24 months), both groups showed significant improvements in clinical outcomes based on AOFAS scores, foot and ankle ability measures (FAAM), and VAS, although the PRP group outperformed the microfracture-only group. "

These are the only details provided by the authors.

34· Line 418: "Curr Rev Musculoskelet Med (2018) 11:616– 

623" should be correct. 

Response: thanks for your comment.

We are sorry for the Mistake; it has been suppressed.

 35· Table IV: outcomes should be better reported.

Response: thanks for your comment.

The outcomes in table 5 have been rewritten according to the outcome reported in different studies hope you are satisfied with our work. Because we needed space, the table was modified, few words needed to be abbreviated to create space to write the outcomes.

36· Lines 543-545: references should be added. Diabetic 

foot ulcers can also be treated by different surgical 

techniques. For example, minimally invasive metatarsal 

osteotomies are applied to treat plantar diabetic foot 

ulcers. It should be added.

Response: thanks for your comment.

The reference was added as requested, but it was obvious that we were talking about the same studies as you requested. It has been added and written in red font in the manuscript.

"Singh et al. [162] combined the results of these and other studies in a systematic evaluation and meta-analysis. The authors concluded that PRP exceeded corticosteroids in VAS and AOFAS scores at three months but showed no difference in pain or function at 1, 6, or 12 months of follow-up[162]."

*Regarding DFU treatment, we added some information as you requested and directly added references, which are written in red in the manuscript.

 "Based on the 22 clinical guidelines available, the current standard of care for treating diabetic foot involves a combination of pressure, shoes, adjunctive therapy, vascular assessment, wound off-loading, infection and glycemic control[8-10], and most importantly, various surgical debridement techniques such as minimally invasive metatarsal osteotomies applied to treat plantar diabetic foot ulcers[12]."

37· Summary should be better explained. 

Response: thanks for your wonderful comment.

As requested, our team reformulated the conclusion, giving more detail about what has been said in the present manuscript, challenges and prospects, especially What needs to be remembered.

We are very glad you have reviewed our article. All your comments help us to improve this article. We hope you are satisfied with our work.

Reviewer 2 Report

The authors have described te use of PRP in ankle and foot related applications.

In the introduction, it would be important to state that the Leukocyte rich PRP preparations can actually enhance inflammation and the ionic balance as described in:  https://doi.org/10.3390/ijms20030721

It would also be useful to state that the effect may be because PRP can help in the viability of bone tissue as investigated in: 10.4172/2329-9509.1000205

Specific opinion:

Line 40 Autologous conditioned plasma is ACP, not PRP

The style of the numbering and coloring of the tables need to be unified as much as possible, and the dimensions need to be added where possible, e.g. time (days) follow-up (days or months?)

Table V: what does centrifugation time mean?

Diabetic foot ulcer is not really a musculoskeletal related condition rather a wound healing and diabetes related issue.

And it would be useful to state in the summary that the non-standardized production method of PRP creates huge differences and comparing the outcomes can be challenging.

The Reference style also needs to be unified.

Author Response

Author's Reply to the Reviewer comments (Reviewer 2 ) 

Response: We appreciate the Editor's and Reviewers' constructive remarks on our work, which have strengthened it. We responded to the criticisms shown on the attached pages, and the changed text in the manuscript is highlighted in red. We hope the revised work will be accepted for publication in the Journal of Clinical Medicine. 

suggestions

1. In the introduction, it would be important to state that the Leukocyte rich PRP preparations can actually enhance inflammation and the ionic balance as described in: https://doi.org/10.3390/ijms20030721

2. It would also be useful to state that the effect may be because PRP can help in the viability of bone tissue as investigated in: 10.4172/2329-9509.1000205

Response: thanks for your suggestion.

As you suggested, we added in the introduction that platelets could enhance inflammation; as we all know, PRP is full of platelet, and so is LR-PRP. This was written in red fond in the manuscript as fallow.

"Additionally, various molecules and features of PRP, such as antibacterial, analgesic, and anti-inflammatory actions, may be advantageous for bone tissue regeneration [15-17]. Moreover, it has been suggested that platelets may activate peripheral blood mononuclear cells (PBMC), which then secrete IL-10, an anti-inflammatory cytokine implicated in tissue regeneration [18]. Leukocyte-rich PRP and leukocyte-poor PRP have been the subject of unresolved disputes for the past several years. However, these and other aspects must be considered when determining a PRP product's optimal biological activity[6]. Despite PRP's beneficial effects on tissue regeneration, its efficacy in bone healing remains debatable [15,20-22]. In vitro studies have demonstrated a dose-dependent effect of PRP on osteoblasts and fibroblast differentiation, with the best outcomes obtained with a low platelet concentration as opposed to a high one [15,20].In contrast, the results obtained from in vivo and clinical research are contested, and variable platelet concentrations of PRP employed, different protocols, the kind of bone defect and different animal species have been suggested as possible causes [15,21,23-25]. PRP utilized as an adjuvant to bone grafts was reported to have a favourable effect in the treatment of periodontal intraosseous deficiencies; however, it appeared to be ineffective in increasing bone growth in sinus lift treatments [22,26].In contrast, PRP injection proved beneficial in surgery for patients with delayed bone union and non-union [1]. "

Specific opinion: 

1.Line 40 Autologous conditioned plasma is ACP, not PRP

Response: we thank you for your great observation.

 The introduction was rewritten as requested by reviewer 1 for a better understanding. So this part has been cancelled from the present manuscript.

2.The style of the numbering and coloring of the tables need to be 

unified as much as possible, and the dimensions need to be 

added where possible, e.g. time (days) follow-up (days or 

months?) 

Response: thanks for your observation.

Text style numbering and colouring have been unified, and relevant dimensions have been corrected.

Our team choose to report the follow-up time using months or weeks, depending on the studies. A slight modification was brought as requested.

3. Table V: what does centrifugation time mean? 

Response: thanks for your comment.

By centrifugation time, we meant the "number of centrifugation", and this mistake has been modified and written in red font in the present manuscript. As we all know, PRP preparation needs centrifugation, and the number differs depending on the producer.

4. Diabetic foot ulcer is not really a musculoskeletal related 

condition rather a wound healing and diabetes-related issue. 

Response: thanks for your comment.

We always have diabetic patients coming to the orthopaedics department for diabetic-related issues.

As you mentioned upward, our team Agreed to change that little subtitle to"Efficacy of PRP Injections in wound healing and diabetes-related issue" written in red font in the manuscript.

5.And it would be useful to state in the summary that the non

standardized production method of PRP creates huge 

differences and comparing the outcomes can be challenging.

Response: we are thankful for this good remarque.

Our team tried to reformulate the conclusion to make it clearer and more concise and insert the information you requested.

6. The Reference style also needs to be unified

Response: thanks for your comment.

Our team worked on that, and we tried our best to unify it as much as possible. The references have all been unified using the reference manager logical and crossref.

Round 2

Reviewer 1 Report

No additional comments.

Reviewer 2 Report

The manuscript has improved and can be accepted now.